# Reduction in Severe, Chronic Mid-Back Pain Following Correction of Sagittal Thoracic Spinal Alignment Using Chiropractic BioPhysics^®^ Spinal Rehabilitation Program Following Prior Failed Treatment: A Case Report with 9-Month Follow-Up

**DOI:** 10.3390/healthcare13202587

**Published:** 2025-10-14

**Authors:** Kyle Longo, Jason W. Haas, Paul A. Oakley, Deed E. Harrison

**Affiliations:** 1Private Practice, St. Louis, MO 63129, USA; kylelongodc@hotmail.com; 2CBP Non-Profit, Inc., Windsor, CO 80550, USA; 3Private Practice, Newmarket, ON L3Y 8Y8, Canada; docoakley.icc@gmail.com; 4CBP Non-Profit, Inc., Eagle, ID 83616, USA; drdeed@idealspine.com

**Keywords:** mid-back pain, thoracic kyphosis, radiography, spine mensuration

## Abstract

We present the findings of a case showing an improvement in severe, chronic mid-back pain (MBP) and disability following sagittal correction of the thoracic spine using Chiropractic BioPhysics^®^ (CBP^®^) spinal rehabilitation with a nine-month long-term follow-up. A 40-year-old female had suffered for years and was referred for spinal rehabilitation by her physicians and physical therapist to treat her severe, chronic MBP. The symptoms had not improved despite several months of physical therapy, traditional chiropractic spinal manipulation, and pain management trigger point injections. The pain was reported as severe and rated as 8/10 at worst on the numerical rating scale. The pain was severe enough to interfere with her normal activities including martial arts training. Postural analysis revealed increased thoracic flexion and spine hyperkyphosis. Lateral thoracic radiography showed a previously undiagnosed wedged vertebral body at T6. Mensuration of the radiograph found an increase in overall posterior tangent angulation from T3–T10 measuring 66.2°. Negative sagittal balance measured from a vertical of T3 above T10 was −16.3 mm. Treatment included Chiropractic Biophysics^®^ (CBP^®^) orthopedic rehabilitation protocols including postural and radiographic based Mirror Image^®^ (MI^®^) exercises, spinal manipulation, and traction. The patient was treated in-office 37 times over the course of 3 months and all initial subjective and objective outcomes were re-assessed. It was reported that the initial average pain of 8/10 for the mid-back had nearly resolved and was rated as 2/10. All ADLs were reported as pain free, including intense exercise and martial arts. Post-treatment radiography was taken following a 24 h “rest-period” and found reduction in the overall hyperkyphosis from T3–T10 now measured 45.2°. Due to the presence of the wedge vertebra, it was recommended that the patient continue home traction and exercises, and long-term follow-up was assessed at 9 months including a repeat of all initial examinations, for subjective and objective outcomes. Thoracic kyphosis was maintained at 47.7° and VAS was 0/10 at 9-month follow-up and symptoms remained nearly resolved.

## 1. Introduction

Conditions involving the thoracic spine can have significant consequences for multiple systems and frequently have pain and dysfunction associated with these postural and structural abnormalities. Musculoskeletal conditions and back pain are consistently found in the top ten causes of disability and this disability and societal and financial burden are tremendous. Pain and dysfunction caused by back pain is frequently the largest burden in lost productivity and disability and impacts all societies across the globe.

A 40-year-old female was referred for corrective chiropractic care by her physical therapist (PT) for severe, chronic mid-back pain (MBP) that had not improved despite several months of PT, traditional chiropractic adjustments, and pain management trigger point injections [1]. The patient reported a history of constant, severe, chronic MBP for over 10 years that worsened with time. She rated her pain as 8/10 at its worst and 6/10 on average on a numeric rating scale (NRS) of 0–10 where 0 is no pain and 10 is max pain. She reported that the pain had caused her to cease certain activities of daily living (ADL) and extended exercise activities, such as martial arts.

The patient agreed to receive treatment that has been previously reported in biomedical literature [2]. This treatment protocol involves the use of mathematical spine and postural models [3]. These models allow for an understanding of the patient’s abnormal postural and spine position and uses specific therapeutics prescribed to lessen the abnormal loads. Previous investigations have shown consistent spine angulation changes coupled with reductions in dysfunction, pain, and worsening disability [4,5,6,7]. Lessening loads on the spine and paraspinal tissues should reduce abnormal loads, lessen aberrant nociception, and lessen the likelihood of future worsening of pain, disability, and degeneration. This study is important because it adds to the literature supporting the conservative treatment of spine pain and associated conditions.

Previous studies of conservative rehabilitation have investigated numerous conditions including pain throughout the spine; however, the specific protocol of prescribed structural exercises, structural traction, and postural spinal manipulative procedures is limited concerning wedge vertebrae in the thoracic spine causing hyperkyphosis, pain, and dysfunction in an athlete and high-functioning martial artist. The reduction in thoracic pain from a relatively simple protocol to duplicate clinically is important for improving the understanding of spine pain and treatment for clinicians, researchers, and institutions. The severity of pain and clinical difference following the unique protocol adds to an understanding of the potential treatment of wedged vertebrae with failure of traditional treatment options.

We present the case of a single patient who reported successful reduction in pain, measured on both the short- and long-term evaluations with visualized reduction in hyperkyphosis and improvement in outcome measures. We aim to expand the literature concerning the potential efficacious impact of this particular therapeutic protocol. This case adds to the biomedical literature and possible treatment options for musculoskeletal pain which is a major contributor to global burden of disease and disability rates [8,9,10,11,12].

## 2. Methods

### 2.1. Ethical Considerations and Declarations

Due to the retrospective nature of the study institutional review board approval is not applicable in agreement with Common Rule’s exemption 45 CFR 46.104. Informed consent for diagnostics and treatment were signed. Patient consent to publish was acquired prior to publication and the report does not contain any identifiable personal data and thus the Declaration of Helsinki is not applicable for this case report. This analysis was conducted with a commitment to ethical integrity and data was used and presented in a way that poses no risk to any individual and intends to improve understanding of spine rehabilitation and pain therapeutics.

### 2.2. Patient History, Examination Results, and Subjective and Objective Findings

The patient received a comprehensive examination including a complete history of the pain and her health conditions. She related that the pain had been progressively getting worse. She reported prior spinal manipulation, massage, trigger point injections, stretching, and general physical therapy to improve strength helped her pain, but only minimally and temporarily. Her history included intense exercise at times, including martial arts training with a whole-body, intense fighting situation technique, and she was able to complete them without pain that interfered with her ability to practice. However, her recent history found that the MBP was beginning to interfere with more of her verbally reported ADLs and was inhibiting her from being as healthy and fit as she was prior to the pain and disability. A 2-way visual analog scale (VAS) was used to assess pain.

Initial orthopedic tests were negative for all regions with only some cervical compression testing increasing her pain in the mid-back, and cervical compression testing without any rotation was the only positive test. The patient was fit, reported no prior medication use beyond OTC pain relievers with minimal benefit, and she was not working at the time of treatment. Visual range of motion (ROM) testing for pain provocation and not magnitude of angulation found thorax flexion and extension to increase mid-back pain. The mid-back paraspinal musculature was tender to palpation for pain provocation. Posture analysis revealed increased sagittal thoracic flexion (hyperkyphosis). Spine radiographs were obtained and the radiographs were measured using PostureRay^®^ Version 26 spine parameter measurement software (PostureCo^®^ Inc., Trinity, FL, USA). Lateral thoracic radiographic examination showed a wedged vertebra at T6, an absolute rotational angle from T3 to T10 (ARA ^T3–T10^) measuring 66.2° (ideal is an elliptical 44°) [13], and sagittal translation (in the *z*-axis) from T3 to T10 (T_z_^T3–T10^) measuring −16.3 mm (ideal is 0 mm) [14] (Figure 1A). All radiographs were obtained with standard neutral positioning, and in accordance with the American College of Radiology guidelines. Measurements of the films were performed with artificial intelligence (AI)-assisted software technology to improve accuracy and reproducibility of the measurements.

### 2.3. Treatment Protocol

She was treated 37 times in-office over 3 months. Each treatment was approximately 1 h in length and daily re-assessment of pain and any other changes were recorded and documented in a standard subjective, objective assessment and plan (SOAP) format. Reportage emphasized the rehabilitation both in-office and at-home care. The treatment was performed in a facility and supervised by a physician and chiropractor, and the reports were shared with the referring PT. The physician determined following the initial examination process that there were no complications from psycho-social issues or other conditions that would be referred to a different specialist. The musculoskeletal nature of the pain pattern coupled with the interference with verbally reported ADLs easily satisfied the necessity for radiography of the spine. The patient reported a significant reduction in her pain from a very early part of the treatment and was consistent with her therapy. The patient received thoracic spinal manipulation (SMT) and Chiropractic BioPhysics^®^ (CBP^®^) Mirror Image^®^ (MI^®^) adjustments using a chiropractic drop table and instrument adjusting using an Erchonia^®^ Adjustor^TM^ instrument (Erchonia Corporation, Fountain Inn, SC, USA). Figure 2A,B demonstrate the supine and prone position of the patient for the drop table, and instrument postural adjustments [4,5,6].

The MI therapies involve positioning the patient in a corrected or over-corrected postural position and addressing different tissues of the spine and posture including muscles, tendons and ligaments, neuromuscular tissues, and PNS and CNS tissues, as well as bone misalignment. In the supine position, the patient had a 3.5″ foam block placed under the apex of the kyphosis in the thorax with the pelvis flat on the bench (Figure 2A). In the prone position, the patient had a 3.5″ foam block placed under her anterior superior iliac spines (ASIS) with the thorax translated anteriorly (+T_z_^T^) and extended (-R_x_^T^) to reduce posterior thorax translation and thoracic flexion (Figure 2B). Sagittal MI mechanical traction was performed in a supine position using the 3D Denneroll^TM^ Traction Table System (Denneroll^TM^ Spinal Orthotics, New South Wales, Australia) with a Thoracic Denneroll^TM^ Spinal Orthotic (Denneroll^TM^ Spinal Orthotics, New South Wales, Australia) placed at the apex of the thoracic hyperkyphosis inducing +T_z_^T^ and -R_x_^T^. As the patient’s tolerance increased, a posterior weighted pull (7 lbs.) of the head was applied to increase thoracic extension and spinal distraction and improve sagittal balance (Figure 2C). A harness with soft padding on the chin and forehead attached to a strap with the weight was used. The patient was able to tolerate 20 min of MI^®^ traction by the third week of care. MI^®^ therapeutic exercises included supine thoracic foam rolling exercises on the floor with a semi-flexible roll used to alleviate muscular stiffness and induce some hysteresis of the paraspinal muscles and ligaments prior to traction. The patient tolerated the traction and therapies very well and was very compliant with the treatment protocol. This protocol has been published previously; however, the uniqueness of each patient requires the application be modified depending on the specific radiographic and postural findings [4]. The treatment protocol increased in duration as the patient was able to tolerate longer times in traction and able to perform more exercises. The postural SMT was consistent across the treatments.

## 3. Results

### 3.1. Post-Treatment Subjective and Objective Findings

A post-treatment exam was performed following 37 CBP^®^ MI spinal rehabilitation visits over 3 months. The same physician who did the initial assessments re-produced the initial tests and recorded the findings. The patient reported high satisfaction with the treatment outcome. She was highly encouraged by the pain relief results and felt her posture had improved as well. The patient stated that her MBP was now occasional, and she rated her pain as 4/10 at its worst and 2/10 on average [15]. This reduction in pain satisfies the minimum clinical expected improvement in pain from the initial exam to the follow-up. She stated she resumed all ADLs including moderately intense exercise, with little if any pain reported and frequently no pain at all. Post-treatment radiographic analysis revealed improvement in sagittal thoracic posture. No orthopedic, neurological, or pain provocation testing was positive for pain and visual ROM for pain provocation found no positions that increased pain. Cervical compression testing was negative. The patients’ compliance to the program and her home care likely influenced her positive outcomes

### 3.2. Post-Treatment Radiographic Findings and Home Care Recommendations

Following the treatment regimen, the patient was instructed to not perform any home exercises or traction and no in-office therapies including MI traction for at least 24 h prior to the post-treatment evaluation. The patient completed VAS and subjective reports and the same physician who acquired the initial radiographs re-produced the initial imaging and the results were measured. Post-treatment lateral thoracic radiographic examination showed improvement in ARA T3–T10 to 45.2°; however, T_z_^T3–T10^ moved posterior to −31.6 mm (Figure 1B). These results demonstrated a significant improvement in postural sagittal and radiographic alignment (Table 1).

The patient was informed that the wedged shape of her T6 vertebrae would lend itself to regression of her thoracic hyperkyphosis, and she understood the need for ongoing supportive therapy, home exercises, and postural awareness. At-home exercises and traction were prescribed for the patient including MI^®^ mechanical traction using the Denneroll^TM^ posture regainer compression extension system unit (Denneroll^TM^ Spinal Orthotics, NSW, Australia) and therapeutic exercise that reproduced the in-office therapies. The patient continued supportive in-office treatments at an average of one day per week and she reported performing home traction 3–4 days per week for 2 months after which she reduced at-home MI mechanical traction frequency to an average of one day per week.

### 3.3. Nine-Month Post-Treatment Long-Term Follow-Up

Following an intermission of frequent in-office treatment, the patient returned for an evaluation after a nine-month break in care. She reported continued home exercises and home traction. However, she had not performed any of her therapeutics within the prior 24 h before the evaluation. A long-term follow-up exam duplicating the initial subjective and objective examinations was performed. The evaluation was completed by the same physician and the patient stated that her MBP continued to be nearly resolved. Her minimal pain had been maintained as very occasional and very mild and worse only after fatigue and with rare frequency. She stated she was able to successfully perform all ADLs including exercise, usually with no pain at all and any discomfort was minimal, and no dysfunction/disability was reported. Pain was rated 2/10 at worst and 0/10 on average (Table 2). Long-term follow-up posture analysis revealed maintained improvement in sagittal thoracic posture. The physician who acquired and measured initial radiographs duplicated the initial radiographic evaluation with mensuration of the spine parameters. PostureRay^®^ digitization software was used for measurement for consistency across platforms. The results of the spine measurement found long-term follow-up lateral thoracic radiographic examination showing sustained improvement in ARA^T3–T10^ to 47.7° and improvement in T_z_^T3–T10^ to −11.2 mm (Figure 1C).

This table reports the visual analog pain scale results at initial evaluation, post-treatment follow-up and 9-month long-term follow-up. 0 is considered no pain and 10 is worst pain possible. The patient reported severe 8/10 pain at the initial assessment that had improved by 50% post-treatment and was reduced to 2/10 at worst at long-term follow-up. Average pain was measured moderate-to-severe 6/10 at initial evaluation and was resolved at long-term follow-up.

## 4. Discussion

We presented the results of conservative treatment for chronic MBP with hyperkyphosis in a 40-year-old female with measured increase in thoracic curve. The patient improved following the initiation of treatment and improvement toward baseline and toward normalization of subjective and objective outcome measures which continued at post-treatment examination as well as 9-month follow-up. The patient reported not only improvement in pain and dysfunction but also improvement in posture and ability to perform pain-free ADLs. There is a significant lack of studies demonstrating conservative therapies impacting MCID improvements in pain and reduced thoracic hyperkyphosis in the medical literature. This study intends to add to the existing evidence for conservative treatment of spine and postural abnormalities causing pain with MCID changes and a non-invasive technique [16].

Thoracic kyphosis angulation has been studied extensively since the introduction of radiography which can allow measurement of the thorax and visualize the thoracic spine segments. Visualization of the thoracic spine from T3 to T10, or recently with digital radiography, T1 to T12, and the ability to accurately diagnosis has greatly improved. Recent studies showing the PostureRay^®^ system can predict accurate spine mensuration at the level of R^2^ = 1 statistical analysis of Harrison Posterior Tangent Method (HPTM) [17,18], indicating the reliability of the radiography assessment is approaching perfect when using the AI software. Advanced and progressively improving AI measurement of spine parameters will push better and more accurate diagnosis, improved treatment recommendations, triage, and differential diagnoses. Previously studied objective outcome measures report and have shown that reducing abnormal spine balance and improving the sagittal spine balance and curvature of the thoracis spine is a desirable clinical outcome [19,20,21]. Increased thoracic spine flexion and progressive translation from a vertical axis line can lead to adult spine deformity (ASD). Reducing ASD in those suffering from hyperkyphotic thoracic spines and progressive pain and worsening disability is a desirable outcome [22,23,24].

Simple radiographic imaging and measurement with cut-points for normal angles vs. pain and disability is readily available globally and AI-assisted mensuration can instantly provide physicians with valuable information regarding referrals for more diagnoses or opinion of surgeons. Thoracic spine surgery for spine misalignments is reported in the literature; however, there are significant risks and very few with long-term improvement and return to ability to perform ADLs without any pain or dysfunction [25,26,27].

The treatment in this case involved the use of a multi-modal spine rehabilitation protocol that is relatively easy to implement in rehabilitation, physical medicine, and chiropractic facilities. High-quality treatment choices which are easily implemented could provide spine treatment providers with a protocol to improve patient outcomes. Protocols which have none of the side-effects of pharmacologic and injection-based medical interventions, as the patient in this report, could benefit others who have tried common treatments. Potential contributions to the patients’ improvement could be explained by the improvement in neuromuscular control and strength of the postural muscles following the treatment.

Trigger point injections (TPIs) of the patient occurred prior to the therapy and were at the direction and recommendation of her medical doctor. She reported the injections consisted of a numbing agent injected into trigger point or tight and tender muscle fibers. The evidence for the use of TPIs is limited and no consensus has been reported of the most effective protocol [28]. The patient reported she had received the TPI treatment many months prior and felt no residual improvement at the time of the initial examination. She reported that the beneficial effects of the TPIs were minimal and short lived, and the procedure was not comfortable enough to warrant long-term usage.

The protocol and analysis of CBP^®^ rehabilitation applied to treat pain has been published prior to this investigation and systematic reviews of literature for the methods of mensuration [29,30,31]. Multiple randomized controlled trials investigating the statistical improvements following treatment as well as with long-term follow-up assessment have shown benefit across multiple health conditions. Biomechanical anatomy modeling studies have further elucidated the foundations of this whole-body rehabilitation program [32,33].

The use of radiography has been shown to be safe and adds to the beneficial understanding of the patient’s unique spine position and the MI^®^ protocols necessary to lessen the abnormal loads [29,30]. Spine radiography offers minimal to no downside, and the risk–benefit use of radiography shows significant benefit in diagnosis and treatment [34,35,36,37]. MI^®^ traction involves visco-elastic structures such as spinal ligaments translated and rotated in the opposite position [38,39]. This traction is held for extended periods to cause stresses and strain forces to change the ligaments toward normal position. This slow stretch prevents injury that can occur at higher loads [40,41,42,43]. This application of MI^®^ positions extends to the neuromusculoskeletal tissues that respond to repetitive activation and perturbations through specific directional exercises [44,45]. CNS regions that control balance and posture are stimulated via afferent mechanoreception while in the MI^®^ postural adjustments. The improvements in latency and amplitude of CNS regions associated with posture have been established in the biomedical literature [46,47,48,49,50,51,52]. Although the literature is limited concerning conservative correction of thoracic hyperkyphosis, a prior study by Oakley et al. found similar results as our current case report [53]. The methods described and used in this study are relatively simple to implement and the ubiquitous nature of radiography in healthcare facilities makes the diagnosis and treatment of conditions such as hyperkyphosis much easier to treat than invasive procedures such as surgery.

Traditional physical therapy, chiropractic SMT, exercise, and TPIs have shown success with temporary reduction in pain; however, there is a significant lack of evidence for both pain reduction and structural improvements with long-term follow-up. Although this is a single case study, it has consistency with previous studies of CBP^®^ rehabilitation. Spine pain reduces quality of life, decreases function, and significantly increases risk of short- and long-term disability. Having simple conservative treatment options is of significant benefit to the population of those suffering daily.

The patient was chosen for potential case report publication due to the uniqueness of the radiographic presentation, the severity of pain, the patients’ significant reduction in quality of life because she could not exercise or perform her chosen sport of martial arts, and the improvements she reported following treatment. The patient was also chosen for publication due to the apparent failure of prior, different conservative and invasive treatments. Following significant changes in radiographs, subjective and objective outcome measures, ADLs, and patient-reported pain demonstrating a minimally clinically important difference (MCID) in 2-way NRS after the treatment protocol, the authors chose to publish the findings. The home care compliance was monitored with a simple chart made by the patient noting the frequency and duration of home traction. No adverse events or side-effects of the treatment were reported by the patient.

Limitations of this study are the single nature of a case report in making definitive conclusions. Another limitation is the lack of additional outcome measures including ADL changes following treatment, which would make the conclusions more robust. The relatively short period between post-treatment and follow-up could limit conclusions and a longer-term follow-up of 2–5 years may show regression toward baseline or continued improvement over time. Future studies could match MBP patients to normal controls without any pain or dysfunction, for studies that measure and compare acute MBP with chronic and normal patients.

## 5. Conclusions

CBP^®^ corrective chiropractic spinal rehabilitation corrected thoracic hyperkyphosis with wedged vertebrae in a 40-year-old female with resumption of intense activities including martial arts without worsening of the condition. Following thoracic spine correction, the patient reported improvement in constant, chronic MBP, disability, and ADL function. This outcome came after other unsuccessful previous interventions (PT for strength and flexibility, traditional chiropractic spinal manipulative therapy, and trigger point injections). Sagittal spinal alignment and posture may result in long-term improvement in MBP, disability, and ADL function, even in trained athletes. This case study shows the need for more research involving spinal rehabilitation of thoracic hyperkyphosis and concomitant health outcome measures, and larger RCTs showing changes in hyperkyphosis and outcome measures.

## Figures and Tables

**Figure 1 healthcare-13-02587-f001:**
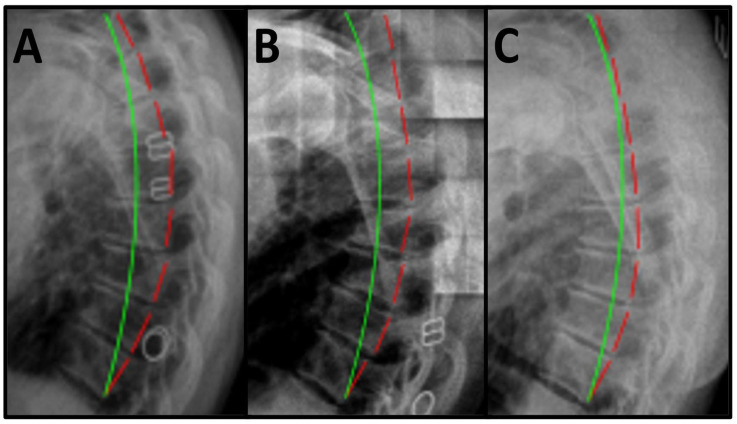
(**A**–**C**). The pre-treatment, post-treatment, and follow-up of the lateral thoracic spine is presented. (**A**), significant flexion and the hyperkyphosis measuring 66.2° from T1–T12 vs. ideal 44° was found. (**B**)—Post-treatment lateral thoracic radiograph demonstrated a reduction in the overall kyphosis to 45.2°. (**C**)—The 9-month long-term follow-up sagittal thoracic radiograph showing continued movement of the spine toward normal and the thoracic kyphosis measured 47.7°. The green line represents the ideal elliptical kyphotic angulation and the red dashed line represents the patient posterior vertebral body. The Harrison Posterior Tangent Method is used to measure segmental and overall kyphosis.

**Figure 2 healthcare-13-02587-f002:**
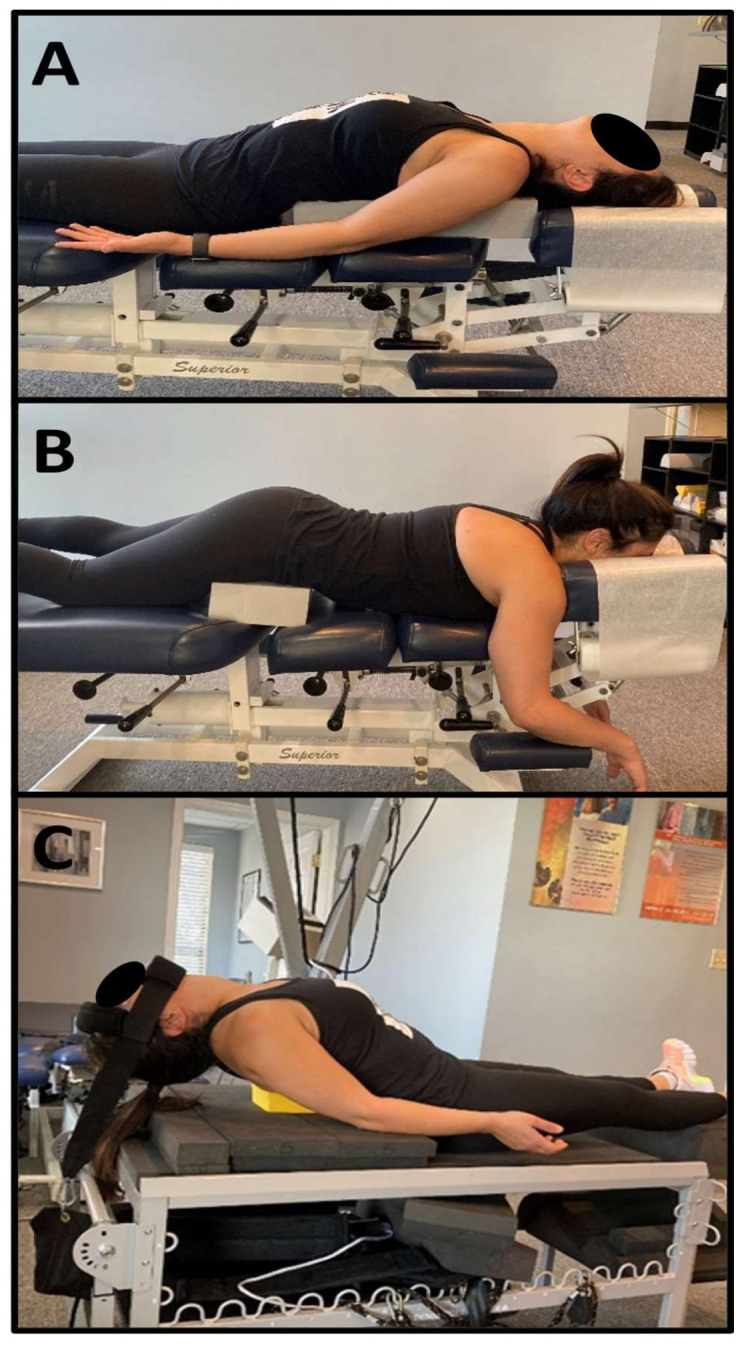
(**A**–**C**). (**A**)—In the supine position, the patient has a rectangular, large foam block under the thoracic spine and the head is allowed to posteriorly translate and extend. The physician used a drop table mechanism which allows the table to assist postural spine MI adjustments where the patient’s head is “pushed” backward. (**B**)—In the prone position, a large foam block is placed under the pelvis, allowing the thorax to anteriorly translate and flatten. The drop table mechanism is used to suddenly “reset” the posture in a position toward normal. (**C**)—The patient is in supine extension MI^®^ traction. A small foam Denneroll^TM^ thoracic orthotic is placed at the peak of the kyphosis. Initially, this was the only traction, but over time, the extension chin strap was used to increase posterior head translation and further flatten the thoracic spine.

**Table 1 healthcare-13-02587-t001:** The pre-, post- and long-term follow-up radiographic findings.

Radiographic Findings	T1–T12 ARA (Ideal is 44°)
Pre-Treatment	66.2°
Post-Treatment	45.2°
Long-Term Follow-up	47.7°

ARA = Absolute Rotation Angle.

**Table 2 healthcare-13-02587-t002:** The 2-way visual analog scale results.

Evaluation	VAS	VAS
Date	Pain at Worst	Average Pain
28 August 2022	8\10	6\10
1 December 2022	4\10	2\10
11 August 2023	2\10	0\10
Overall Change	6\10	6\10

## Data Availability

Data from our manuscript will be made available upon request from the corresponding author Jason W. Haas due redactions that would be necessary to protect patient privacy.

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
