# Peer review of "Reduction in Severe, Chronic Mid-Back Pain Following Correction of Sagittal Thoracic Spinal Alignment Using Chiropractic BioPhysics^®^ Spinal Rehabilitation Program Following Prior Failed Treatment: A Case Report with 9-Month Follow-Up"

_healthcare, 2025, doi:10.3390/healthcare13202587_

Round 1
Reviewer 1 Report
Comments and Suggestions for Authors
Introduction
Context and background
The introduction immediately starts with the description of the patient. It does not first explain the clinical importance of thoracic hyperkyphosis, its prevalence, or the general impact of mid-back pain on public health. Readers should be given more context before diving into the specific case.
Knowledge Gap
It mentions that the treatment "has been previously described" and that the study "expands the literature," but it doesn't clearly specify what is still unknown or what specific treatment was implemented. It would be helpful to include a brief explanation of what previous studies have shown and what remains uncertain – this is the "research gap" the article fills.
Rationale for the case report
While it says the case “adds to biomedical treatment options,” it doesn’t specify why this particular case is worth reporting (e.g., unusual severity, rare failure of prior treatments, unique radiographic findings, or long-term follow-up data).
Objectives or research question
The introduction should clearly state the aim of the case report, such as:
To demonstrate the effectiveness of CBP® spinal rehabilitation in correcting thoracic hyperkyphosis.
To show how this correction translates into pain reduction and functional improvement after failed standard treatments.
Methodology
Patient selection rationale
The report does not explain why this patient was chosen for publication. A statement on the uniqueness of the case (e.g., failure of standard care, unusual radiographic findings, or long-term follow-up) would strengthen the methodological transparency.
Comprehensive patient demographics
Apart from age and sex, other relevant information is missing, such as body mass index (BMI), comorbidities, medications, lifestyle factors, and occupation, all of which could influence spinal health and treatment outcomes.
Diagnostic standards
Although radiographs and posture assessments were performed, the methodology does not specify which diagnostic guidelines were followed or how examiners ensured the accuracy and reproducibility of the measurements.
Outcome measures
Pain was measured using NRS and VAS, but there is no justification for these choices nor mention of additional validated functional scales (e.g., Oswestry Disability Index, Roland-Morris Disability Questionnaire) that could provide a broader perspective on disability and quality of life.
Treatment standardization
The description of CBP® interventions is detailed, but it is unclear whether the protocol strictly followed a published standard or was tailored to this patient. This limits reproducibility.
Monitoring adherence to home care
The methods mention home traction and exercises, but do not explain how patient compliance was assessed or whether adherence influenced outcomes.
Adverse events reporting
There is no systematic description of how potential risks or side effects of treatment were monitored or documented.
Data analysis
The report presents descriptive results, but the methodology does not specify how improvements were defined (e.g., minimum clinically important difference for pain scales) or whether changes were compared to normative reference values.
Results
Structured presentation of outcomes
The results are written in a descriptive way, but they lack a clear, structured format that separates baseline, post-treatment, and follow-up data in a consistent manner. Tables and figures could better highlight changes over time.
Statistical or clinical significance
Improvements are described, but there is no reference to thresholds for minimal clinically important differences (MCID) in pain or postural correction, which would indicate whether the observed changes are clinically meaningful.
Documentation of adverse events
The results focus exclusively on improvements but do not state explicitly whether any adverse events, setbacks, or treatment-related discomforts occurred during therapy or follow-up.
Patient perspective
Although the patient is reported to be “highly satisfied,” a more formalized measure of satisfaction or quality of life would add credibility (e.g., patient-reported outcome measures).
Discussion:
Stronger link to the literature
The discussion cites prior studies but does not fully compare this case with similar published cases or clinical trials. A deeper comparison would clarify how this case aligns with, extends, or challenges existing evidence.
Mechanistic explanation
While the text mentions spinal alignment and load reduction, the physiological and biomechanical mechanisms underlying pain reduction could be explained in more depth (e.g., neuromuscular adaptation, ligamentous remodeling, CNS involvement).
Author Response
Context and background
The introduction immediately starts with the description of the patient. It does not first explain the clinical importance of thoracic hyperkyphosis, its prevalence, or the general impact of mid-back pain on public health. Readers should be given more context before diving into the specific case.
Thank you for your comment. We added some description of hyperkyphosis and pain consequences in the introduction
Knowledge Gap
It mentions that the treatment "has been previously described" and that the study "expands the literature," but it doesn't clearly specify what is still unknown or what specific treatment was implemented. It would be helpful to include a brief explanation of what previous studies have shown and what remains uncertain – this is the "research gap" the article fills.
Thank you for your comment. We added a brief explanation to the introduction bridging the research gap concerning conservative treatment and why the prior studies may have gaps in explanation of specific treatment for specific conditions such as hyperkyphosis.
Rationale for the case report
While it says the case “adds to biomedical treatment options,” it doesn’t specify why this particular case is worth reporting (e.g., unusual severity, rare failure of prior treatments, unique radiographic findings, or long-term follow-up data).
Thank you, we expanded the introduction to include this recommendation.
Objectives or research question
The introduction should clearly state the aim of the case report, such as:
To demonstrate the effectiveness of CBP® spinal rehabilitation in correcting thoracic hyperkyphosis.
To show how this correction translates into pain reduction and functional improvement after failed standard treatments.
Thank you for the comment. We clarified and expanded the introduction to include this information.
Methodology
Patient selection rationale
The report does not explain why this patient was chosen for publication. A statement on the uniqueness of the case (e.g., failure of standard care, unusual radiographic findings, or long-term follow-up) would strengthen the methodological transparency.
Thank you for the comment. We expanded the methodology section to include a statement as to why the lead author chose to attempt publication of the patient’s findings.
Comprehensive patient demographics
Apart from age and sex, other relevant information is missing, such as body mass index (BMI), comorbidities, medications, lifestyle factors, and occupation, all of which could influence spinal health and treatment outcomes.
Thank you for your comment. We added clarification to the methods section.
Diagnostic standards
Although radiographs and posture assessments were performed, the methodology does not specify which diagnostic guidelines were followed or how examiners ensured the accuracy and reproducibility of the measurements.
Thank you for the comment. We added a statement concerning the technique and measurement of the radiographs for clarity.
Outcome measures
Pain was measured using NRS and VAS, but there is no justification for these choices nor mention of additional validated functional scales (e.g., Oswestry Disability Index, Roland-Morris Disability Questionnaire) that could provide a broader perspective on disability and quality of life.
Thank you for your comment. We understand that the case could demonstrate more significant outcomes if these measures were taken, however, the physician did not obtain additional outcome measures prior to treatment.
Treatment standardization
The description of CBP® interventions is detailed, but it is unclear whether the protocol strictly followed a published standard or was tailored to this patient. This limits reproducibility.
Thank you for the comment. We added a statement concerning the reproducibility of the protocol from previously published studies.
Monitoring adherence to home care
The methods mention home traction and exercises, but do not explain how patient compliance was assessed or whether adherence influenced outcomes.
Thank you. We clarified the patient home care compliance and commented on the potential benefit from this compliance.
Adverse events reporting
There is no systematic description of how potential risks or side effects of treatment were monitored or documented.
Thank you for the comment. We added a statement regarding potential adverse events and possible side effects of the treatment.
Data analysis
The report presents descriptive results, but the methodology does not specify how improvements were defined (e.g., minimum clinically important difference for pain scales) or whether changes were compared to normative reference values.
Thank you. We clarified the MCID improvement in pain from the initial assessment to the post-treatment and long-term follow-up.
Results
Structured presentation of outcomes
The results are written in a descriptive way, but they lack a clear, structured format that separates baseline, post-treatment, and follow-up data in a consistent manner. Tables and figures could better highlight changes over time.
Thank you for the comment. We clarified the results section.
Statistical or clinical significance
Improvements are described, but there is no reference to thresholds for minimal clinically important differences (MCID) in pain or postural correction, which would indicate whether the observed changes are clinically meaningful.
Thank you for your comment. We added a statement regarding the MCID improvement in pain following treatment.
Documentation of adverse events
The results focus exclusively on improvements but do not state explicitly whether any adverse events, setbacks, or treatment-related discomforts occurred during therapy or follow-up.
Thank you for your comment. We added a clarification statement regarding adverse events and potential side effects.
Patient perspective
Although the patient is reported to be “highly satisfied,” a more formalized measure of satisfaction or quality of life would add credibility (e.g., patient-reported outcome measures).
Thank you for your comment. Future studies will be sure to include more PROs and Outcomes measuring HRQoL. This study included a 2 way NRS and subjective observation and reports from the patient. We strengthened the patient reported subjective outcomes throughout the manuscript. If there is a specific line number, please allow us to further revise.
Discussion:
Stronger link to the literature
The discussion cites prior studies but does not fully compare this case with similar published cases or clinical trials. A deeper comparison would clarify how this case aligns with, extends, or challenges existing evidence.
Thank you for the comment. We expanded and clarified the discussion to address how our case report adds to the limited literature concerning conservative correction of hyperkyphosis and pain reduction.
Mechanistic explanation
While the text mentions spinal alignment and load reduction, the physiological and biomechanical mechanisms underlying pain reduction could be explained in more depth (e.g., neuromuscular adaptation, ligamentous remodeling, CNS involvement).
Thank you for your comment. We expanded the discussion to include these potential contributions to the patient’s positive outcome.
Reviewer 2 Report
Comments and Suggestions for Authors
This manuscript presents a case report of a 40-year-old female patient with severe chronic mid-back pain associated with thoracic hyperkyphosis. The patient had undergone several conventional treatments, including physical therapy, chiropractic adjustments, and trigger point injections, all of which failed to provide lasting relief. Following a course of CBP rehabilitation, the patient experienced substantial pain reduction and functional improvement, which were sustained at 9-month follow-up.
However, several areas require clarification:
- As a single case report, the conclusions should be presented more cautiously. Some statements should be rephrased on more balanced terms.
- A 9 month follow up is valuable but still short for spinal deformity outcomes.
- While the figures show improvements, the reliability of the measurements (intra/inter rater variability) should be discussed. Were the radiographs assessed independently or blinded?
- Some of the cited references are authored by the CBP research group. Independent studies and broader literature on non-surgical thoracic spine interventions should be included for balance.
- The manuscript needs to more clearly highlight what is unique about this case compared to prior CBP literature (e.g., presence of a wedge vertebra, long-term functional recovery in martial arts). Without this, the report risks appearing redundant.
- Statements such as “no risk and only benefit” regarding radiography are overly strong and may be viewed as biased. A more cautious, balanced interpretation of X-ray follow-up and its potential risks is needed.
- While some disclosures are included, the strong affiliation with CBP Non-Profit requires an explicit limitations section that acknowledges potential bias and emphasizes the need for independent replication.
- Several citations are over 15 years old. Adding more recent meta-analyses or RCTs would strengthen the discussion.
The manuscript provides a successful case report demonstrating pain and postural improvement with CBP treatment following the failure of other interventions. With improvement, it will be much better.
Author Response
- As a single case report, the conclusions should be presented more cautiously. Some statements should be rephrased on more balanced terms.
Thank you for your comment. We have modified the conclusion to report the conclusions in a more balanced fashion.
A 9 month follow up is valuable but still short for spinal deformity outcomes.
Thank you for your comment. We plan on writing a longer term follow up if the patient is available. Unfortunately at the time of writing, the patient is no longer available for longer-term follow-up. CBP researchers strive to include the longest possible long term follow-up, some as long as 13 years. We will strive to obtain follow-up if the patient (who is not at the same address) is available in the distant future and would submit a long term follow-up irrespective of the outcomes measured at that time.
While the figures show improvements, the reliability of the measurements (intra/inter rater variability) should be discussed. Were the radiographs assessed independently or blinded?
Thank you for the comment. The radiograph assessment utilized PostureRay software. This software has an inter and intra reliability co-efficient that reaches R2=1 for AI to AI reliability and is incredibly accurate for AI to human reliability. We cited this study in reference #16: Hosseini, M.M.; Mahoor, M.H.; Haas, J.W.; Ferrantelli, J.R.; Dupuis, A.-L.; Jaeger, J.O.; Harrison, D.E. Intra-Examiner Reliability and Validity of Sagittal Cervical Spine Mensuration Methods Using Deep Convolutional Neural Networks. J. Clin. Med. 2024, 13, 2573. doi: 10.3390/jcm13092573Sd
And stated in the article, “Visualization of the thoracic spine from T3-T10, or recently with digital radiography, T1-T12 ability to accurately diagnosis has greatly improved. Recent studies showing the PostureRay® system can predict accurate spine mensuration at the level of R2= 1 statis-tical analysis of Harrison Posterior Tangent Method (HPTM) [16].”
We added a statement in the revised manuscript to further elucidate the reliability of the measurements.
Some of the cited references are authored by the CBP research group. Independent studies and broader literature on non-surgical thoracic spine interventions should be included for balance.
Thank you. References 7, 10, 11, 19, 21, 22, 23, 26 all discuss non-surgical, surgical, rehabilitative and medicinal treatments for mid back pain with some referencing hyperkyphosis specifically. Unfortunately, there is a dearth of specific studies involving radiography, wedged vertebrae, high pain levels, and successful other than CBP studies. We were unable to find successful radiographic studies previously reported.
The manuscript needs to more clearly highlight what is unique about this case compared to prior CBP literature (e.g., presence of a wedge vertebra, long-term functional recovery in martial arts). Without this, the report risks appearing redundant.
Thank you for the comment. We clarified the patient selection and uniqueness of the case.
Statements such as “no risk and only benefit” regarding radiography are overly strong and may be viewed as biased. A more cautious, balanced interpretation of X-ray follow-up and its potential risks is needed.
Thank you for your comment. We clarified the statement.
While some disclosures are included, the strong affiliation with CBP Non-Profit requires an explicit limitations section that acknowledges potential bias and emphasizes the need for independent replication.
Thank you for your comment. We provided an extensive COI statement to the journal.
Several citations are over 15 years old. Adding more recent meta-analyses or RCTs would strengthen the discussion.
Thank you for your comment. Unfortunately, the literature is limited in regard to successful conservative therapies for hyperkyphosis and treatment of wedged vertebrae.
Reviewer 3 Report
Comments and Suggestions for Authors
This is a well-written single case report with a good therapeutic outcome.
However, there are a few concerns and remarks.
- What is unique and innovative about this case to justify its publication?
- The results should be compared with previous similar reports. For example, Oakley et al 2018 have reported a series of patients with hyper kyphosis treated with CBP MI
- Some clinical data are missing. More detailed clinical signs , diagnostics' results such as osteoporosis in view of the wedged vertebral body, past medical history etc
- Diagram does not actually show the angle, but elliptical lines
- ADL is not measured by a scale. Because of the retrospective nature of the study, this should be mentioned as a limitation.
- More details on the scientific rational of the advantages of the method over other methods should be provided
- A take-away lesson from this case is not clear, other that CBP can be a useful method
- Suggestions for future studies should include clinical trials, what is the purpose of matching MBP patients to normal controls?
Author Response
This is a well-written single case report with a good therapeutic outcome.
However, there are a few concerns and remarks.
- What is unique and innovative about this case to justify its publication?
Thank you for your comment. We clarified and expanded the uniqueness of the case selection and knowledge gap bridged by the case report.
- The results should be compared with previous similar reports. For example, Oakley et al 2018 have reported a series of patients with hyper kyphosis treated with CBP MI
Thank you for your comment, we referenced this study and commented on the similar findings.
- Some clinical data are missing. More detailed clinical signs , diagnostics' results such as osteoporosis in view of the wedged vertebral body, past medical history etc.
Thank you for the comment. We expanded the methods section to include more of this information.
- Diagram does not actually show the angle, but elliptical lines
Thank you for the comment. The computer software PostureRay uses the posterior tangents to form the angle. This is represented by the dashed red lines on the radiographs. The green elliptical line is the ideal thoracic model.
- ADL is not measured by a scale. Because of the retrospective nature of the study, this should be mentioned as a limitation.
Thank you for your comment. We added this as a limitation to the study.
- More details on the scientific rational of the advantages of the method over other methods should be provided.
Thank you for the comment. We expanded the advantages of conservative treatment vs other methods.
- A take-away lesson from this case is not clear, other that CBP can be a useful method.
Thank you for your comment. We expanded the uniqueness of the case report as well as CBP as a viable treatment option for similar conditions.
- Suggestions for future studies should include clinical trials, what is the purpose of matching MBP patients to normal controls?
Thank you for the comment. Matching conditions with normal controls is a very common practice for observational studies and RCTs. The matching of normal controls with patients suffering from pain is a frequent design element in larger studies.
Reviewer 4 Report
Comments and Suggestions for Authors
Dear Editor and Authors,
First of all, I would like to thank you for the opportunity to review this study. I would like to highlight some limitations that, in my opinion, should be addressed and revised.
Introduction
-
It would be advisable to include some lines on the epidemiology of mid-back pain (MBP).
-
The introductory section should not discuss the success or failure of the therapy applied, but rather provide the justification and objective of the case report, both of which should be included.
Methodology
-
Please specify the orthopedic tests used to evaluate the range of motion (ROM).
-
Which scale was employed to assess activities of daily living (ADLs)? Please include this scale along with the necessary references for each assessment method used for the different variables.
-
Modify the caption of Figure 1ABC to improve clarity, as part A appears only with the letter, whereas B and C are accompanied by “Figure.”
-
The treatment protocol section again mentions that the patient improved; however, such information should be presented in the Results section, not here.
-
The treatment protocol should be presented more clearly and concisely. What was the frequency, duration, or intensity of the treatment? Was it consistent across all weeks? Please provide this information in a schematic format.
Results
-
Pain measurements are reported, but in the Methodology section there is no description of the assessment protocol or scale employed, which is mandatory.
-
You state: “Post-treatment posture analysis revealed improvement in sagittal thoracic posture.” Where are the corresponding data that support this improvement?
-
I recommend completing the tables with pre- and post-intervention values for each variable, as well as specifying the measurement time points, which are currently unclear. The addition of a graphical representation could also be considered.
Discussion
-
This section should be reformulated, as it does not provide evidence or hypotheses based on the scientific literature to explain why this treatment might have had the positive effects described.
-
Please expand the limitations section (e.g., the evaluator was the same pre- and post-intervention, the absence of additional functional assessments, and the inclusion of only a single patient as the sample size).
Conclusion
-
The conclusion should be reformulated, as it is too extensive and written in a categorical tone. The findings may suggest potential improvements but cannot be confirmed or extrapolated to the wider population.
Author Response
First of all, I would like to thank you for the opportunity to review this study. I would like to highlight some limitations that, in my opinion, should be addressed and revised.
Introduction
- It would be advisable to include some lines on the epidemiology of mid-back pain (MBP).
Thank you for the comment. We expanded the introduction and discussion to include this information.
- The introductory section should not discuss the success or failure of the therapy applied, but rather provide the justification and objective of the case report, both of which should be included.
Thank you for the comment. We modified the introduction to expand the justification and objective of the study.
Methodology
- Please specify the orthopedic tests used to evaluate the range of motion (ROM).
Thank you for the comment. We clarified the methods section to discuss the visual ROM analysis for pain.
- Which scale was employed to assess activities of daily living (ADLs)? Please include this scale along with the necessary references for each assessment method used for the different variables.
Thank you for the comment. Unfortunately, no objective measure was used to determine ADLs. Patient reporting of improved ability was verbally reported. We clarified this in the manuscript.
- Modify the caption of Figure 1ABC to improve clarity, as part A appears only with the letter, whereas B and C are accompanied by “Figure.”
Thank you for the comment. We changed and clarified the figure explanation.
- The treatment protocol section again mentions that the patient improved; however, such information should be presented in the Results section, not here.
Thank you. We clarified and modified the methods and results sections to clarify based on your comment.
- The treatment protocol should be presented more clearly and concisely. What was the frequency, duration, or intensity of the treatment? Was it consistent across all weeks? Please provide this information in a schematic format.
Thank you for your comment. We expanded the methods section to discuss the treatment protocol. For brevity, the authors have chosen not to add a schematic diagram of the protocol.
Results
- Pain measurements are reported, but in the Methodology section there is no description of the assessment protocol or scale employed, which is mandatory.
Thank you. We added this to the methods section.
- You state: “Post-treatment posture analysis revealed improvement in sagittal thoracic posture.” Where are the corresponding data that support this improvement?
Thank you. We clarified this in the results.
- I recommend completing the tables with pre- and post-intervention values for each variable, as well as specifying the measurement time points, which are currently unclear. The addition of a graphical representation could also be considered.
Thank you for your comment. A table was added to the results section.
Discussion
- This section should be reformulated, as it does not provide evidence or hypotheses based on the scientific literature to explain why this treatment might have had the positive effects described.
Thank you for your comment. The discussion was expanded to provide this information.
- Please expand the limitations section (e.g., the evaluator was the same pre- and post-intervention, the absence of additional functional assessments, and the inclusion of only a single patient as the sample size).
Thank you for your comment. The limitations were expanded to include this information.
Conclusion
- The conclusion should be reformulated, as it is too extensive and written in a categorical tone. The findings may suggest potential improvements but cannot be confirmed or extrapolated to the wider population.
Thank you for the comment. The conclusion was re-written.
Round 2
Reviewer 1 Report
Comments and Suggestions for Authors
Thank you for implementing the suggested changes. The presented Case Report, in its current form, can be accepted.
Author Response
Thank you for implementing the suggested changes. The presented Case Report, in its current form, can be accepted.
Thank you for your review.
Reviewer 3 Report
Comments and Suggestions for Authors
The addition in yellow '' The patient's compliance to the program .... the authors chose to publish the findings'' added to justify the uniqueness of the case contains a lot of redundant arguments and does not belong to the Results Section but to the Discussion Section. Please justify the presentation of this work as an illustrative case report that demonstrates the clinical utility of the CBP method in this specific clinical scenario in a highly -competent athlete and move this to the Discussion. Also, the point asking for the scientific rationale behind the method asked you to explain why the CBP method might have been more efficient than other conventional physiotherapy methods previously applied to this patient, not about overall conservative treatment vs surgery. The manuscript also needs minor syntactic language editing.
Author Response
The addition in yellow '' The patient's compliance to the program .... the authors chose to publish the findings'' added to justify the uniqueness of the case contains a lot of redundant arguments and does not belong to the Results Section but to the Discussion Section. Please justify the presentation of this work as an illustrative case report that demonstrates the clinical utility of the CBP method in this specific clinical scenario in a highly -competent athlete and move this to the Discussion. Also, the point asking for the scientific rationale behind the method asked you to explain why the CBP method might have been more efficient than other conventional physiotherapy methods previously applied to this patient, not about overall conservative treatment vs surgery. The manuscript also needs minor syntactic language editing.
Thank you for your comment. We moved some of the information from the results to the discussion. We added a why statement clarifying how recent studies of PT or SMT or TPIs may improve pain short term, there are few long term structural and functional improvements. We made some additional syntactic language editing.
Reviewer 4 Report
Comments and Suggestions for Authors
no further comments
Author Response
Thank you for your review.